# γ-Valerolactone Production from Levulinic Acid Hydrogenation Using Ni Supported Nanoparticles: Influence of Tungsten Loading and pH of Synthesis

**DOI:** 10.3390/nano12122017

**Published:** 2022-06-11

**Authors:** Gerardo E. Córdova-Pérez, Jorge Cortez-Elizalde, Adib Abiu Silahua-Pavón, Adrián Cervantes-Uribe, Juan Carlos Arévalo-Pérez, Adrián Cordero-Garcia, Alejandra E. Espinosa de los Monteros, Claudia G. Espinosa-González, Srinivas Godavarthi, Filiberto Ortiz-Chi, Zenaida Guerra-Que, José Gilberto Torres-Torres

**Affiliations:** 1Laboratorio de Nanomateriales Catalíticos Aplicados al Desarrollo de Fuentes de Energía y Remediación Ambiental, Centro de Investigación de Ciencia y Tecnología Aplicada de Tabasco (CICTAT), DACB, Universidad Juárez Autónoma de Tabasco, Km.1 Carretera Cunduacán-Jalpa de Méndez, Cunduacan CP 86690, Tabasco, Mexico; enrique_cordova90@hotmail.com (G.E.C.-P.); link-190@hotmail.com (J.C.-E.); adibab45@gmail.com (A.A.S.-P.); adrian.cervantes@ujat.mx (A.C.-U.); carlos.arevalo@ujat.mx (J.C.A.-P.); adrian.cordero@ujat.mx (A.C.-G.); alejandra.espinosa@ujat.mx (A.E.E.d.l.M.); 2Investigadoras e Investigadores por Mexico, Universidad Juárez Autónoma de Tabasco, División Académica de Ciencias Básicas, Centro de Investigación de Ciencia y Tecnología Aplicada de Tabasco (CICTAT), Km.1 Carretera Cunduacán-Jalpa de Méndez, Cunduacan CP 86690, Tabasco, Mexico; clauidia.espinosa@gmail.com (C.G.E.-G.); godavarthi.srinivas@gmail.com (S.G.); filiberto.ortiz@ujat.mx (F.O.-C.); 3Tecnológico Nacional de México Campus Villahermosa, Laboratorio de Investigción 1 Área de Nanotecnología, Km. 3.5 Carretera Villahermosa–Frontera, Cd. Industrial, Villahermosa CP 86010, Tabasco, Mexico; zenaida.gq@villahermosa.tecnm.mx

**Keywords:** hydrogenation, levulinic acid, y-valerolactone, tungsten, Ni/Al_2_O_3_-TiO_2_ nanocatalysts

## Abstract

γ-Valerolactone (GVL) has been considered an alternative as biofuel in the production of carbon-based chemicals; however, the use of noble metals and corrosive solvents has been a problem. In this work, Ni supported nanocatalysts were prepared to produce γ-Valerolactone from levulinic acid using methanol as solvent at a temperature of 170 °C utilizing 4 MPa of H_2_. Supports were modified at pH 3 using acetic acid (CH_3_COOH) and pH 9 using ammonium hydroxide (NH_4_OH) with different tungsten (W) loadings (1%, 3%, and 5%) by the Sol-gel method. Ni was deposited by the suspension impregnation method. The catalysts were characterized by various techniques including XRD, N_2_ physisorption, UV-Vis, SEM, TEM, XPS, H_2_-TPR, and Pyridine FTIR. Based on the study of acidity and activity relation, Ni dispersion due to the Lewis acid sites contributed by W at pH 9, producing nanoparticles smaller than 10 nm of Ni, and could be responsible for the high esterification activity of levulinic acid (LA) to Methyl levulinate being more selective to catalytic hydrogenation. Products and by-products were analyzed by ^1^H NMR. Optimum catalytic activity was obtained with 5% W at pH 9, with 80% yield after 24 h of reaction. The higher catalytic activity was attributed to the particle size and the amount of Lewis acid sites generated by modifying the pH of synthesis and the amount of W in the support due to the spillover effect.

## 1. Introduction

At the beginning of the 21st century, one of humanity’s challenges is facing is the production of energy as a consequence of the growing world demand due to the increase in the population. The depletion of oil, one of the main sources of energy, and the impact of CO_2_, which is one of the greenhouse gases causing climate change, are some of the factors that make the development of renewable and environment friendly sources of energy one of the most important challenges of the modern world. All these have led to the expansion of new sources of sustainable energy that protect the environment, such as the use of biomass for this purpose [1]. Lignocellulosic biomass is a promising raw material for the production of high added-value chemicals and biofuels because it is abundant and renewable, and as it is normally a non-used waste it does not belong to any food chain [2]. Levulinic Acid (LA) is a rising platform molecule according to the US Department of Energy [3,4]. Due to its two functional groups (ketone and carboxylic acid) it can be transformed into various high added-value chemicals such as γ-Valerolactone (GVL), 1,4-pentanediol, succinic acid, 3-hydroxypropanoic acid, and 2-methyltetrahydrofuran [5]. LA can be produced by hexoses, which are the key components of cellulose. Catalytic dehydration of hexoses or cellulose results in the formation of 5 HMF [4,6], and through hydration it can form LA [7]. On an industrial scale, several companies around the world have developed processes for the production of levulinic and furfural acid, focusing on viable commercial applications of levulinic acid, such as lactones, levulinate esters, or valeric biofuels [8,9]. Most of these processes involve GVL as an intermediary. LA hydrogenation results in GVL as a product, which is a stable and low-toxic molecule of great application due to its unique physicochemical characteristics, high boiling points (207–208 °C) and flash point (96 °C), low vapor pressure, and air inertia [10,11]. GVL applications are based on the production of liquid biofuels, solvents, food additives, aromatics, or oxygenated gasoline additives [12,13,14,15]. GVL is considered a better alternative to ethanol as a fuel additive, because it has a significantly lower vapor pressure and a higher energy density compared with ethanol [16]. Bruno et al. tested the properties of the blend fuel GVL/gasoline and found that the addition of GVL led to a considerable decrease in CO emissions [17].

GVL can be obtained by hydrogenation of LA or by alkyl levulinate using heterogeneous catalysts based on noble metals such as Ru, Au, Ir, Rh, Re, Pd, and Pt [18,19,20,21,22]. Upare et al. reported a 98.6% yield of GVL on Ru/C catalyst using dioxane as a solvent at high H_2_ pressures, while when using Pd/C and Pt/C catalysts the yields of GVL were 90% and 30% respectively, so Ru showed a better activity [23]. Yan et al. studied the liquid phase hydrogenation of LA to GVL using methanol as a solvent with a Ru/C catalyst of (5%) with selectivity of 99% to GVL and a LA conversion of 92% in a batch reactor (130 °C, 1.4 Mpa H_2_) in methanol. However, the activity and stability of the Ru/C catalyst was not reproducible due to active metal leaching [24]. Lange et al. also reported a serious problem of leaching/deactivation of active metals of the heterogeneous catalyst in LA hydrogenation. Although carbon supports overcome the leaching problem to some extent, they do not allow regeneration of the deactivated catalyst [25].

Despite Ru’s excellent catalytic activity under mild reaction conditions, the main challenges are high costs and leaching that can limit its application on an industrial scale.

To avoid using noble metal catalysts, Ni-based catalysts have a relatively higher stability during the reaction as advantages and they can be easily recycled due to their magnetism, making them an alternative for this type of reaction. Mallesham et al. [26] documented that when using the (30%)Ni/SiO_2_ catalyst a selectivity of GVL >97% was achieved with a LA conversion of 54.5% during 20 h of reaction.

Fu et al. [27] observed that by using a catalyst (40%)Ni/Al_2_O_3_ prepared by wet impregnation, a selectivity of 99% of GVL was achieved with a LA conversion of 100% in dioxane during 4 h of reaction (180 °C and 30 bar H_2_). Although dioxane improves the stability of nickel catalysts, the use of this solvent is not recommended as it is carcinogenic [28]. Jiang et al. [29] studied Ni/MgO-Al_2_O_3_ in different ratios of Mg/Al and Ni/MgAlO_2.5_, obtaining an optimal GVL yield of 99.7% (160 °C, 30 bar H_2_). The larger surface area enhanced nickel dispersion on the MgO-Al_2_O_3_ support, explaining the high activity and selectivity towards GVL compared with Ni/MgO and Ni/Al_2_O_3_. On the other hand, Lv et al. [30] obtained optimal results (100% conversion of LA and 93.3% selectivity of GVL) using Ni/MgO from a series of prepared catalysts (Ni/SiO_2_, Ni/Al_2_O_3_, Ni/TiO_2_, Ni/ZrO_2_, and Ni/ZnO) using 2-propanol as solvent and a donor of H_2_ at 150 °C for 2 h. It has been reported that metal leaching can be suppressed during the reaction by using alcohols as solvent [31,32].

Al-Shaal et al. [33] studied the influence of solvents (methanol, ethanol, 1-butanol, 1,4-dioxane, and mixtures of metanol-H_2_O, ethanol-H_2_O, and butanol-H_2_O). Among the alcoholic solvents used, methanol showed the highest LA conversion and GVL yields. This observation was attributed to the high solubility of H_2_ in the solvent compared with others used in this study. On the other hand, the use of primary alcohols promotes the esterification of LA. Enumula et al. [34] studied the influence of WO_3_ in the catalyst (3%)WO_3_/SBA-16 boosting the esterification of LA with ethanol, methanol, propanol, and butanol. Within these solvents, methanol and ethanol achieved the highest selectivity of 96% and 95% respectively.

Meanwhile, Kumar et al. [35] studied the catalyst (20%)Ni/TiO_2_ with a conversion of LA of 68.8% and a selectivity to GVL of 88.8%, with (1.5%)WO_3_(20%)Ni/TiO_2_; the Brönsted sites generated by WO_3_ were responsible for opening the GVL ring producing valeric acid (VA) decreasing the selectivity of GVL. Mafokoane et al. [36] obtained similar results using Cuo/Al_2_O_3_-WO_3_ (6%) as the catalyst, obtaining a higher conversion of LA, but the selectivity of GVL decreased due to the increase in Lewis-type acidity, creating 2-Methyltetrahydrofuran (2-MTHF). Yuan et al. [37] reported a 100% ring opening and decalin conversion using Ni-WO_3_/Al_2_O_3_-Zeolite at 345 °C, attributed to abundant strong Brönsted acid sites with an increase in WO_3_ loading until reaching the 18% by weight. In addition to the importance of the particle size for the hydrogenation of LA to GVL, the acid-base properties and the structural characteristics of the support represent an important role at the moment of hydrogenation [38].

It has been found that the moderate Lewis-–Brönsted acidity in the support is responsible for the dehydration of the 4-hydroxy pentanoic acid important intermediate in the production of GVL [39]. A study using Montmorillonite (MMT) showed how strong acidity facilitates LA esterification and cyclization, while Ni metal sites promoted selective hydrogenation of LA to GVL [31]. Enumula et al. [40] found that in the ZrO_2_/SBA-15 catalyst the acidity increased with an increase in the percentage of ZrO_2_ (25% by weight), improving the selectivity to GVL. However, a further increase in acidity up to 30 wt% ZrO_2_ led to the generation of undesired products due to GVL ring opening, which produced a decrease in GVL selectivity.

LA conversion using Ni/Al_2_O_3_, Cu/Al_2_O_3_, and Ni-Cu/Al_2_O_3_ at 250 °C and 6.5 MPa H_2_ demonstrated a rapid deactivation of the Ni/Al_2_O_3_ catalyst caused by a greater formation of carbon on the surface. The NiAl_2_O_4_ catalyst has been reported to provide greater resistance to carbon formation. Relatively large particle sizes > 10 nm facilitate carbon formation [41,42], not only it providing better efficiency in hydrogenation by spillover effect, but also resistance to carbon deposition.

Hengst et al. [43] studied various synthesis methods and the (5%)Ni/Al_2_O_3_ was the best catalyst prepared by the wet impregnation modified with urea, showing a 90% LA conversion and 75% GVL yield (140 °C and 30 bar H_2_) using dioxane as solvent for 4 h. The use of urea showed a larger surface area besides obtaining a particle size of 7 nm, which were important factors in the catalytic activity. In fact, several studies emphasized the importance of smaller particle sizes <10 nm, which promote better performance in the hydrogenation of LA or its alkyl levulinates to GVL [44,45,46].

Table 1 shows different Ni-based monometallic supported catalysts and experimental conditions selected for the synthesis of the GVL using LA as the platform molecule. These previous studies evaluated different loadings of Ni and types of supports showing catalytic performance effects.

The catalyst support is key in Figure 1, which shows the different reaction routes. Route 1 is the hydrogen reduction of the carbonyl group to obtain an intermediate 4-hydroxylevulinic acid (4-HPA) followed by intramolecular esterification to obtain GVL. Route 2 is carried out by the esterification of LA by the alcohol functional group followed by hydrogenation and subsequently dealcoholization, promoting GVL formation. Route 3 involves dehydration of LA to Angelica Lactone (α/β-AL) and after that hydrogenation of the double bond C=C to give GVL [46]. Oxides such as Al_2_O_3_, ZrO_2_, TiO_2_, and their mixtures TiO_2_-ZrO_2_, Al_2_O_3_-TiO_2_-W, and WO_3_-Ta_2_O_5_ have been tested in biomass decomposition, obtaining good results [6,53,54,55,56].

Hun et al. [57] showed that Al_2_O_3_ works as a good acid support due to the formation of Brönsted (-OH) and Lewis (Penta coordinated Al^3+^) acid sites that can promote dehydration which can occur during GVL synthesis, according to path 1 and 3.

Improved catalytic activity in LA conversion and selectivity to GVL has been related to well-dispersed metal particles and their size along with the combination of Lewis acid-base sites. Nowadays, due to the high price of noble metals and metal leaching, the production of GVL has been replaced with the use of transition metals such as Fe, Cu, Co, and Ni [58].

Consequently, the novelty of the present work falls in several points: (1) we used Ni as a transition metal, (2) we used methanol as a reaction medium due to its high selectivity to Alkyl Levulinate (AL), which is an important intermediate in the production of GVL [34], as can be seen in path 2, (3) we synthesized Ni using a suspension impregnation method that has not been reported before and has led to generation of particle sizes lower than 10 nm, and (4) we used a complex three-component support (Al_2_O_3_-TiO_2_-WO_3_), varying the W content and the PH of the medium synthesis of supports, creating variability of acidity levels of the support and reducibility of metal that is beneficial for GVL yield.

In fact, it has been proved that support is able to modify the properties of the active phase such as its reducibility, dispersion, and stability due to a closed intimate interaction between the metal particles and the support. The support plays an important role to obtain a better dispersion of the metallic phase associated with the lattice defects or structural defects in nanomaterials, such an oxygen vacancies or unsaturated sites. Additionally, the modification of one oxide support using the doping of support enhances more the activity, selectivity, and stability of the catalysts [53,54]. The strong metal support interaction is beneficial to avoid metal leaching [59,60].

Therefore, this work studied the addition of W in the Ni/Al_2_O_3_-TiO_2_-WO_3_ catalyst to different pH values of support synthesis and its effect on the amount of Lewis acid sites and the Ni particle size in the GVL yield. Methanol was used as the reaction medium because there is little information on the conversion of LA to GVL. Considering all the above, we presented a series of Ni/Al_2_O_3_-TiO_2_-WO_3_ catalysts with a load of 10% Ni and 1%, 3%, and 5% W as a bifunctional active catalyst in the conversion of LA to GVL and its esterification of methyl levulinate at 170 °C and 4 MPa of H_2_ in a batch reactor during 24 h of reaction. The by-products were analyzed by H^1^ nuclear magnetic resonance. The catalysts were prepared using the suspension impregnation method and characterized by nitrogen physisorption, adsorbed pyridine by Fourier transform infrared spectroscopy (FT-IR), X-ray diffraction (XRD), UV-vis spectroscopy with diffuse reflectance (DRS UV-Vis), temperature-programmed reduction of hydrogen (TPR-H_2_), temperature-programmed desorption of hydrogen (TPD-H_2_), Scanning Electron Microscopy (SEM), Transmission Electron Microscopy (TEM), and X-ray Photoelectron Spectroscopy (XPS).

## 2. Materials and Methods

### 2.1. Reagents

All reagents used were of analytical quality and were used without any purification treatment: Levulinic acid (Sigma-Aldrich, Toluca city, Mexico, 99.5% purity), Gamma-valerolactone (Sigma-Aldrich, 99.5% purity), Angelica-lactone (Sigma-Aldrich, 99.5% purity), Methanol (Meyer, 99.9% purity), Aluminum trisecbutoxide C_12_H_27_AlO_3_ (Sigma-Aldrich, 97% purity), Titanium butoxide (IV) Ti[O(CH_2_)_3_CH_3_]_4_ (Sigma-Aldrich, 97% purity), hydrated ammonium metatungstate (NH4) (NH_4_)_6_H_2_W_12_O_40_·xH_2_O, nitrate nickel (II) hexahydrate Ni(NO_3_)_2_·6H_2_O, n-butanol (99.9%, Baker), and Sec-butanol (Sigma-Aldrich, 99.5% purity). All experiments were performed using ultrapure water (18.2 MΩ cm^−1^) from a PureLab model Option-Q water purifier.

### 2.2. Catalyst Support Synthesis

Sol-Gel method was used for the synthesis of the Al_2_O_3_-TiO_2_-WO_3_ support, modifying the pH to 3 and 9 using acetic acid [CH_3_COOH] and ammonium hydroxide [NH_4_OH] respectively. The Al_2_O_3_/TiO_2_ ratio was 50/50 by weight, while the percentage of W was 1%, 3% and 5% by weight. The synthesis was carried out in aqueous medium, using 1-butanol [CH_3_-(CH_2_)_3_-OH] and 2-butanol [CH_3_CH_2_CH(OH)CH_3_] as solvent with ultra-pure water to conduct the hydrolysis between the butoxide titanium [Ti(OCH_2_CH_2_CH_2_CH_3_)_4_] and the aluminum tri-sec-butoxide Al[OCH(CH_3_)C_2_H_5_]_3_ (Sigma-Aldrich, 97% purity). The water/alcohol molar ratio was 1/8 by volume and alkoxide/water was 1/16. The alcohol/alkoxide mixture was added to a flask with constant stirring until a homogeneous mixture was obtained. Once the mixture was homogenized, the water was added slowly over 3 h. The pH of the water was adjusted to pH 3 and pH 9 respectively and, immediately afterwards, the hydrated ammonium metatungstate salt ((NH_4_) _6_H_2_W_12_O_40_ • xH_2_O) was added until it was completely dissolved. After the dripping, it was left under constant stirring for 24 h at a temperature of 80 °C. Once the gel was formed, the solvent was removed in a rotary evaporator at 80 °C and left to dry in an oven at 120 °C for 48 h. Finally, it was calcined at 500 °C with a heating ramp of 2 °C/min. Table 1 shows the labels of materials.

### 2.3. Ni Supported Catalysts Preparation

Ni catalysts supported on Al_2_O_3_-TiO_2_-WO_3_ were prepared by the suspended impregnation method [61]. The support was heated in a reducing atmosphere with hydrogen flow at 90 mL/min using a heating ramp of 2 °C/min, until reaching a temperature of 300 °C and it remained in this way for 1 h. Subsequently, the support was cooled to a temperature of 25 °C, changing the nitrogen flow for 30 min. Nickel nitrate salt Ni(NO_3_)_2_·6H_2_O was added in 20 mL of water, and calculations were carried out to have a 10% by weight of nickel in relation to the support. It was kept under nitrogen flow for 10 min, and later mixed with the support under nitrogen flow to have a homogeneous mixture. After 30 min, the nitrogen flow was changed to hydrogen. It was brought to a temperature of 80 °C and kept for 12 h in order to evaporate the water. Then, the reduction of the material was carried out in a reducing atmosphere of hydrogen with a flow of 90 mL/min with a ramp of 2 °C/min until reaching 450 °C for 4 h. Finally, the catalyst was cooled to 25 °C in a hydrogen flow, then removed, stored, and labeled. All the catalysts prepared for this study, as well as their labels, are shown in Table 2.

### 2.4. Characterization Techniques

#### 2.4.1. Nitrogen Physisorption

The determination of the specific area, diameter, and pore volume of the catalysts was conducted by the N_2_ physisorption technique. It was performed on equipment of surface area measurement, MICROMERITICS TRISTAR 3020 II at 77 K (−196 °C). To remove impurities, a 0.1 g sample was weighed and degassed for 3 h at 300 °C. Data were analyzed in ASAP 2020 software to determine specific area (Sg), pore volume (Vp), and pore size distribution (PSD) using the B.E.T. method (Brunauer, Emmet, and Teller).

#### 2.4.2. X-ray Diffraction (XRD)

X-ray diffraction analysis was used to determine the composition of the phases and to estimate the crystallite size of the powders. X-ray diffraction (XRD) was performed using a Bruker D2 PHASER diffractometer with Co Kα radiation source (λ = 0.179 nm) during an analysis time of 660 s. The analysis was carried out in the range of 20° to 80°. The JADE 6 database helped to complete the identification of the phase. The average size of the crystals in the catalysts was estimated using the Scherrer equation:(1)$$D=\frac{0.9\;}{\;Cos\theta }$$

#### 2.4.3. Diffuse Reflectance UV-Vis Spectroscopy (DRS UV-Vis)

The UV-Vis Diffuse Reflectance Spectra were performed on a Varian Cary 300 spectrophotometer, in the range of 800 to 200 nm, equipped with an integrating sphere. BaSO_4_ compound with 100% reflectivity was used as a reference.

#### 2.4.4. SEM (Scanning Electron Microscopy)

SEM micrographs of the samples were recorded using a Scanning Electron Microscope (JEOL, USA Inc., Peabody, MA, USA, model JSM-6010LA).

#### 2.4.5. X-ray Photoelectron Spectroscopy (XPS)

The XPS spectra of the samples were recorded using a SPECS^®^ spectrometer with a PHOIBOS^®^ 150 WAL hemispheric energy analyzer with angular resolution (<0.5 degrees), equipped with an XR 50 X-ray Al-Ray and µ-FOCUS 500 X-ray monochromator (Al excitation line). To protect the fresh and spent sample it was transferred to the XPS chamber without exposing it to air (using a mobile XPS chamber under Ar). The binding energy of C 1s (284.8 eV) was used as a reference. Binding energies (B_E_) and intensities for chemical quantitation were determined after subtracting a Shirley-type background from the photoemission spectra using XPS peak 4.1 software.

#### 2.4.6. Pyridine FTIR Analysis

Solid samples were analyzed by FTIR using pyridine as a probe molecule to determine the acid properties of samples following the method according to which the samples are analyzed in the form of self-supporting tablets, previously subjected to in situ activation under vacuum at 400 °C before absorbing pyridine. All the analyses were carried out on a NICOLET FTIR equipment model Magna 560 with a resolution of 4 cm^−1^ and 50 scans, and a DTGS detector.

#### 2.4.7. Temperature Programed Reduction of Hydrogen (TPR-H_2_)

The reduction at programmed temperature of hydrogen (TPR-H_2_) of the monometallic catalysts was completed in Bel Japan Belcat-B equipment equipped with a thermal conductivity detector (TCD). The experiments were carried out using 50 mg of reduced catalyst. It was heat treated in a cell with argon for one hour at 400 °C, using a heating rate of 10 °C/min and an argon flow of 50mL/min. It was cooled to room temperature and passed through a flow of the 5% H_2_/95% Ar mixture. TPR thermograms were recorded using a heating ramp at 10 °C/min with a temperature range of 50 to 900 °C with a flow rate of 10 mL/min.

#### 2.4.8. Temperature Programed Desorption of Hydrogen (TPD-H_2_)

The desorption at programmed temperature of hydrogen (TPD-H_2_) of the monometallic catalysts was completed in a Bel Japan Belcat-B equipment equipped with a thermal conductivity detector (TCD). The experiments were carried out using 50 mg of reduced catalyst. It was heat treated in a cell with argon for one hour at 550 °C, using a heating rate of 10 °C/min and an argon flow of 50 mL/min. It was cooled to 40 °C and passed through a flow of the 5% H_2_/95% Ar mixture of 20 mL/min for 1 h. Finally, a flow of Ar was passed for 1 h (T = 40 C, Ar_flow_ = 50 mL/min). TPD thermograms were recorded using a heating ramp at 10 °C/min with a temperature range of 50 to 500 °C with a flow rate of 50 mL/min of Ar for 45 min.

#### 2.4.9. High Resolution Transmission Electron Microscopy (HRTEM)

TEM analysis was performed using JEM-2100 (JEOL, Tokyo, Japan) equipment operating at an acceleration voltage of 200 kV. The powder was treated with a sonication in isopropanol to ensure a homogeneous dispersion. A small drop was deposited on the carbon films on a 200-mesh copper grid, which was introduced into the TEM Analysis chamber after complete evaporation of the solvent.

### 2.5. Catalytic Tests

All the hydrogenation reactions from LA to GVL were carried out in a stainless-steel autoclave of 50 mL high pressure, equipped with a magnetic stirring system. The reaction was executed using 0.6 g of LA in a solution of 30 mL of methanol, applying 0.2 g of catalyst with a 3:1 ratio (LA/catalyst). The reaction was completed at a temperature of 175 °C, with a stirring of 500 rpm under a 40 bar pressure of H_2_. The reaction was monitored for 24 h, and samples were taken at different times.

#### 2.5.1. Analysis of the LA and GVL after Reaction Tests

##### Gas Chromatographic Analysis (GC)

The reaction crude was filtered after the reaction, when the reactor was cooled to room temperature, to later be analyzed in a Shimadzu GC-2010 Plus gas chromatograph, equipped with a FID detector and a capillary column HP-5 19091J-413 (30 cm × 0.32 mm × 0.25 µm). The initial temperature of the column was 80 °C with a temperature ramp of 10 °C/min, with an injection volume of 0.5 µL; the SPLIT injector temperature was 250 °C and the detector temperature was 270 °C with a flow of 30 mL/min of H_2_ and 300 mL/min of air, and the carrier gas was He with a flow of 25 mL/min.

The conversion of levulinic acid and yield of GVL were calculated using the following equations:(2)$$\%\;\mathrm{GVL}\;yield=\frac{Moles\;of\;\mathrm{GVL}\;produced}{Inicial\;moles\;of\;\mathrm{LA}}$$
(3)$$\%\;\mathrm{LA}\;Conversion=\frac{Moles\;of\;\mathrm{LA}\;reacted}{Inicial\;moles\;of\;\mathrm{LA}}$$

##### Hydrogen Nuclear Magnetic Resonance ^1^H-NMR Analysis

The solvent was evaporated at the reaction crude and subsequently analyzed by ^1^H NMR to identify the products of the conversion from LA to GVL and the by-products generated. A Bruker Advance III 600 MHz NMR spectrometer was employed using deuterated chloroform (CDCl_3_, 25 °C). The spectra were processed with the MestReNova program ^1^H NMR (600 MHz, CDCl_3_).

## 3. Results and Discussions

### 3.1. Catalysts Characterization

#### 3.1.1. N_2_ Physisorption

The form of the isotherms of the ATW A and ATW B supports were analyzed, showing type IV isotherms typical of mesoporous materials (2–50 nm) according to IUPAC [62]. Isotherms in Figure 1a reveal two types of hysteresis in all the supports in both synthesis methods due to the geometry of the titanium having a hysteresis loop H1 (P/P_0_) of 0.7, indicating the large size of the inlet of the pores in a spherical and continuous way because of the interaction of WO_3_ and TiO_2_ as shown by previous studies [36,63,64]. On the other hand, a hysteresis loop of the H2 type, representative of solids with non-uniform cylindrical pores due to the interaction of Al_2_O_3_-TiO_2_, with a bottleneck shape appeared [65,66]; this hysteresis loop was located between a relative pressure (P/P_0_) of 0.8–0.9 in the supports and indicates the pore interconnectivity [67]. The surface area increased with the percentage of WO_3_, obtaining a larger surface area in the supports modified at pH 3 in contrast with the supports at pH 9 (See Table 2). These results are consistent with those reported by another research group, who observed that a greater surface area was obtained using acetic acid to set the pH at 3 [68].

Figure 1b shows the isotherms of the Ni catalysts and that, like the isotherms of the supports, they did not have changes in the type of isotherm with the nickel deposit. This indicates that there was a good thermal stability and despite the reductive treatment at 450 °C, they still have type IV isotherms, characteristic of mesoporous materials. All the isotherms of the supported Ni catalysts presented a hysteresis loop of type H1 and type H2 in a range of (P/P^0^) 0.5–1, which indicates that the porous structure of the materials was maintained. The isotherms showed a decrease due to the obstruction of the pores by the nickel deposit, generating pores of smaller size with respect to the support. According to the results, the stability of the structure is a result of thermal stability, because of the synergy between TiO_2_-Al_2_O_3_ and its interaction with Ni particles. The high percentage of titanium could be a factor, among others, that result in thermal stability of supported Ni catalysts, as mentioned by Escobar et al. [68]; on the other hand, Zhang et al. [42] prepared Ni/Al_2_O_3_ catalysts, preserving the structure and geometry of the catalyst pores, despite the Ni deposit at temperatures above 700 °C; this thermal stability is consequence of the strong interaction between Ni and Al_2_O_3_ in concentration of 35% to 50% *w*/*w* of nickel and at temperatures from 400 °C to 800 °C. The structure is maintained due to the presence of NiAl_2_O_4_, bringing stability and integrity to the structure up to high temperatures. The decrease of a certain percentage (~37%) in the surface area of the Ni/ATW1 A and Ni/ATW3 A catalysts regarding the support is due to the blocking of the pores by the metallic charge. Only a loss of (~14%) was observed for the Ni/ATW5 material in which there was the highest amount of tungsten. As seen in Table 3, greater stability is obtained when the support contains tungsten (5% *w*/*w*). The pore volume decreases, and the pore diameter increases at concentrations < 5%, which could indicate a partial collapse of the pores due to the smaller pore diameter as the effect of pH which, in this case, is 3. Similarly, Li et al. [69] observed that by incorporating Ce on the surface of Al_2_O_3_, the decrease in surface area is related to the good dispersion of the metallic charge on the surface, diffusing into the pores.

#### 3.1.2. X-ray Diffraction (XRD)

Figure 2a,b show the X-ray diffraction results of the Monometallic materials prepared with 10% Ni by the suspension method. Signals are observed at 2θ = 44.5°, 51.8°, and 76.3° corresponding to the planes (111), (200), and (220) respectively. These peaks were identified using the technical sheet JCPD-04-0850, which belongs to the Ni metallic species with an FCC crystalline phase (face-centered cubic), and this metallic Nickel structure is present in all materials as well as in supports prepared at pH 3 (see Figure 2b) and at pH 9 (see Figure 2a). Weak signals were found assigned to the γ-Al_2_O_3_ JCPD 10-0339 phase, the presence of which even after the incorporation of Ni may be due to a phase transition between the γ-Al_2_O_3_ phase and NiAl_2_O_4_ because it is well dispersed on the support. The existence of the NiAl_2_O_4_ spinel crystalline phase would originate diffraction peaks coinciding with those attributed to Al_2_O_3_ with gamma phase. This is because the pseudo-spinel structure of γ-Al_2_O_3_ has lattice parameters very similar to those of NiAl_2_O_4_ spinel, which makes both species indistinguishable. However, a slight displacement of the diffraction peaks at smaller angles and an increase in the intensity of the diffraction peak at 46.8° in contrast with the peak at 66.7° may indicate an enrichment in NiAl_2_O_4_ species versus γ-Al_2_O_3_ [70] and TiO_2_ due to the reducing treatment that was given to the support, causing the dehydroxylation of the surface. The oxygen vacancies generated would induce Ni dispersion and formation of the NiAl_2_O_4_ spinel [71].

According to these considerations, it can be stated that since NiO phase is not detected, and it could be possibly found in nanoparticles, undetectable by this technique, dispersed on a support, part of them reduced to metallic Ni and diffused on the Al_2_O_3_ matrix, forming NiAl_2_O_4_.

#### 3.1.3. Diffuse Reflectance UV-Vis Spectroscopy (DRS UV-Vis)

Figure 3a,b show the results of the UV-Vis DRS spectra of the Ni/ATW catalysts. In the range of 375–450 nm and 700–750 nm, Ni species are related in octahedral coordination (Ni^2+^_oct_) and in the range of 450–700 nm they are associated to a tetrahedral coordination (Ni^2+^_tetra_) due to to its electron configuration of 3d^8^ [72,73,74,75]. Gullapelli et al. [74] used a Ni/Al_2_O_4_ and Ni/TiO_2_ catalyst where the position of the bands at 350–450 nm and 700 nm favored the coordination of Ni^2+^ species; the coordination relationship is based on temperature and on nickel charge. Obtaining NiO at a metal load greater than 20% in *w*/*w* and less than this load favors the formation of NiAl_2_O_4_ according to Gullapelli et al. [74]. The modification of the band at 220–350 nm and 360–380 nm in Ni catalysts is related to NiO in octahedral coordination, and the increase in intensity at 345 nm could be related to the growth of NiO, while on the other hand at 510 nm it is related to ʋ_2_ (^3^A_2g_ → ^3^T_1g_) transitions with an octahedral coordination [76]; this species could not be identified in X-ray diffraction, so this band is associated to the nickel spinel (NiAl_2_O_4_).

As stated by Zurita-Mendez et al. [77] and Scheffer et al. [78], the modification in the 220–350 nm region with respect to the support is due to the charge transfer d–d (^3^A_2_g/^1^T_1_g) of O^2−^ → Ni^2+^ of Ni^2+^_oct_ sites that are associated with NiAl_2_O_4_. The intense region at 450–700 nm was associated with the bands of ʋ_3_ (^3^A_2g_ → ^3^T_1g_ (P)) and ʋ_2_ (^3^A_2g_ → ^3^T_1g_) that resulted from the d–d transitions of Ni^2+^ ions contained in octahedral sites, which could conclude the existence of Ni^3+^ species that the octahedral sites are occupied by Ni^3+^ within nickel aluminate NiAl_2_O_4_. These bands are caused by the absorption of Ni^2+^ in coordination Ni^2+^_tetra_ and Ni^3+^/Al^3+^ in octahedral coordination [72], as confirmed by the results of Ray diffraction and Raman spectroscopy (not shown here). Then again, the effect of W, on the support, is clear due to the band at 370–450 nm that could be the contribution of the interaction of Ni^2+^ coordinated Ni^2+^_oct_ with tungsten species, which can form NiWO_4_. This species has bands characteristics at 280 nm, 370 nm, and 740 nm according to studies by Scheffer et al. [78]; however, the X-ray diffraction results did not identify patterns assigned to this phase.

The intensity of the bands at 500 nm could be related to the oxygen vacancies generated by the reduction of Ti^4+^ to Ti^3+^ as a result of the reduction of Ni^2+^ to Ni^0^ [79]. The increase in the absorption band may be due to the degree of vacancies generated by the Ti^3+^ species, distorting the network and causing defects resulting from the change in oxidation state from Ti^4+^ to Ti^3+^ [80].

#### 3.1.4. TPR-H_2_

Figure 4a shows the reduction profiles of hydrogen consumption of the Ni/ATW1 A catalyst, and it is observed that the reduction of NiO to Ni^0^ was in the range of 300–450 °C; this reduction was carried out in a single step (NiO + H_2_ → Ni + H_2_O), being in a lower proportion and with a smaller particle size since, according to the results found in the literature, it is mentioned that a large particle size requires a higher reduction temperature. Corresponding to the XRD technique, no signals were observed to the NiO crystal structure, which is why they are found in sizes smaller than 3 nm in relation to the detection limit of the technique.

As claimed by Ewbank et al. [81], “free” NiO is that nickel that is found on the surface of the catalyst in the form of nickel oxide at a reduction temperature lower than 450 °C; therefore, in section I located in a temperature range between 300 and 450 °C, the reduction of NiO to Ni is located on the surface of the support, which has a lower interaction. Additionally, the literature mentions the reducibility of Ni species supported on γ-Al_2_O_3_ and how it varies depending on the metallic charge. Zeilnski et al. [82] mentioned how a uniform layer of stoichiometric nickel aluminate is created only on the surface of γ-Al_2_O_3_ when there is a Ni load of 20% by weight. Gao et al. [83] mentioned that the reduction of these species is found in section II located at a temperature between 450 and 750 °C due to a soft interaction with the support, mainly with NiO-Al and NiO-W but with a high content of NiO; this area is associated with the reduction of Ni^2+^ species that are found forming a non-stoichiometric surface aluminate (NiAl_x_O_y_), highly dispersed on the surface of the support [82]. Kumar et al. [84] and Yang et al. [85] reported that these species are reduced at temperatures above 550 °C, so that W, when in contact with Ni, promotes its dispersion and interaction with the support. In the same way, reduction of Ti^4+^ to Ti^3+^ takes place in this area at a temperature of 550 °C generated due to spillover effect, the activation of H_2_ on Ni sites, which homolytically dissociates H_2_ [86,87,88]. For the peak located in section III, they have a strong interaction, so a reduction temperature higher than 700 °C is needed [88], and the least reducible NiO is in the NiAl_2_O_4_ stoichiometry phase, requiring high temperatures for its reduction; the contribution recorded at 750 °C is associated with the reduction of the Ni of spinel NiAl_2_O_4_ in bulk or stoichiometric form; there was no defined peak at lower temperatures because the Ni particles were dispersed by the Ti and W species, which can obstruct the formation of spinel, promoting the dispersion of nickel [35,89].

The profiles obtained from the Ni/ATW5 A catalyst are shown in Figure 4b, and it can be observed that the increase in the percentage of W 1% to 5% of the support makes more evident the displacement of the reduction peaks at higher temperatures compared with the profiles of the monometallic Ni/ATW1 A catalyst. The deconvolution of the reduction profiles of Ni/ATW5 A revealed a proportion of free NiO in the section I, and the reduction peaks in this section decreased as the percentage of W in the support increased; it is possible that Ni is dispersed in areas of TiO_2_ or W [35]. In section II, an increase in the reduction peaks was observed due to a greater interaction of the NiO particles with the support, and the increase in W generated a better dispersion of Ni on the support which promoted the reduction of Ti^4+^ to Ti^3+^ due to to electron transfer from Ni atoms [89], and charge transfer from Ti^4+^ to WO_3_ [64] is also possible. In section III, it is shown how the reduction peaks increased compared with when the support had 1% W, so it is possible that the presence of W during the synthesis of Ni/ATW5 A promoted the rearrangement of the ƴ-Al_2_O_3_ phase and the interaction of Ni, resulting in the formation of NiAl_2_O_4_ which, unlike NiAl_x_O_y_, requires temperatures above 700 °C for its reduction. These results indicate that a temperature of 450 °C is insufficient for complete Ni reduction with 5% tungsten, and that temperatures higher than 700 °C are needed for when the percentage of WO_3_ > 1%; this may be due to the fact that the increase in WO_3_ promotes the stability of the NiAl_2_O_4_ phase [90]. Because of the octahedral and tetrahedral coordination of Ni^2+^, higher temperatures are required for its reduction; according to Horsley et al. [91], only 30% of nickel species are reduced at temperatures of 400 °C with 11% nickel, which suggests that 70% of nickel is in the oxidized state, and the profile shift at higher temperatures with 5% W proposes a reduction of large groups of NiO particles and also that the particles interact with tungsten up to certain point in the reduction of NiO. The shift of the reduction peak towards high temperatures with increasing W loading is probably due to an interaction between tungsten oxide and nickel particles [35].

#### 3.1.5. Temperature-Programmed Desorption of Hydrogen (TPD-H_2_)

Figure 5 shows the TPD-H_2_ profile of Ni/ATW1 A and Ni/ATW5 B catalysts, where a peak located at 120 °C was identified in both catalysts. This peak at low temperature is due to the desorption of H_2_ formed from the H atoms adsorbed on the Ni nanoparticles [92] formed from the reduction of free NiO and non-stoichiometric NiAl_2_O_4_. It can also be associated with the adsorption/desorption of H_2_, which are related to the morphology and different sizes of metallic particles [93]. No other peaks were observed in the profiles at temperatures above 300 °C, the highest desorption occurred at a temperature of 120 °C.

The low dispersion of Ni is due to the few metallic species available on the surface. Salvati et al. [20] mentioned that for an 11% Ni/Al_2_O_3_ catalyst, only 30% of the Ni^2+^ is reduced to the metallic state, and this agrees with the TPR profiles of H_2_ where a strong interaction of Ni with the support was observed.

The increase in W content in the Ni/ATW5 B nanomaterial caused an increase in the intensity of the peak at 120 °C with respect to the Ni/ATW1 A nanomaterial, this result suggesting that it has more available metal sites [92]. Crisóstomos et al. [94] attributed the peaks in a range of temperatures (<400 °C) to the desorption of nickel particles dispersed on the support that come from the reduction of non-stoichiometric NiAl_2_O_4_ that corresponds to a stronger metal–support interaction having resistance to sintering [42].

The metallic dispersion and particle size of Ni/ATW A and Ni/ATW B are reported in Table 4. The dispersion of metallic surface species from Ni/ATW1 A and Ni/ATW5 B was 11.5% and 12.5% respectively, although the Specific Surface Area of 245 m^2^/g and 317 m^2^/g was high. This effect may be due to the tungsten species that are formed using different pH during the synthesis of the supports [95]. These results suggest that the W in the basic medium due to the effect of the hydroxyl groups on the surface could help the incorporation of Ni intro the framework of a support lattice, due to the presence of extra oxygen vacancy sites caused for the addition of another oxide during the support synthesis method, which helps Ni dispersion and consequently decreases the particle size and increases the dispersion, as suggested by Chary et al. [93] using a catalyst with 10% *w*/*w* of Ni supported on Al_2_O_3_.

#### 3.1.6. Scanning Electron Microscopy (SEM)

The micrographs shown in Figure 6 correspond to monometallic catalysts. Larger particles are observed in Figure 6a corresponding to the Ni/ATW1 A catalyst, in contrast to the Ni/ATW5 B catalyst of Figure 6b. This effect is due to the use of NH_4_OH modifying the textural properties, diffusing the nickel in a more effective way within the pores because the volume and average pore diameter are slightly higher, as previously discussed according to the results of physisorption of N_2_, which contributed to a better dispersion of nickel. Higher concentrations of Ni were observed in abundant areas of titanium, which confirms that metallic nickel is mainly found on the surface of titanium and Ni^2+^ is in strong interaction with Al_2_O_3_, forming the spinel as observed in the EDS analysis of backscattered electrons (BSE) where the high contrast in the micrographs is due to the heavier elements of Figure 7 and Figure 8.

In Figure 7a, corresponding to the Ni/ATW1 A catalyst, the results suggest that the elements that compose the support (Al, Ti, and W) were homogeneously distributed. Nickel in Figure 7e is scattered on the surface, both Ti, Al, and W; however, areas of greater accumulation were observed, and the parts of greater contrast are Ni in the metallic state. Zurita et al. [77] suggested that these clear areas are due to NiO particles, but according to the X-ray diffraction results, this phase was not identified, and instead only the Ni and NiAl_2_O_4_ phases were. According to these results, the lighter parts in the micrographs were caused by the Ni metallic species dispersed on the surface of the support as shown in Figure 7b,e. The darker parts were due to Al_2_O_3_ and TiO_2_, on the other hand, the more agglomerated part implied a strong interaction of Ni and Al, producing the formation of the NiAl_2_O_4_ spinel, as suggested by Liao et al. [96] when using Ni/Al_2_O_3_ at 8% by weight Ni; this signified that the Ni^2+^ species have the tendency to join Al_2_O_3_ using percentages below 15% by weight. Gullapelli et al. [74] mentioned that spinel formation could begin at low temperatures, forming a non-stoichiometric NiAl_2_O_4_ spinel at the surface level when Ni^2+^ contacts the support at temperatures below 500 °C. Therefore, the spinel would be found in the monolayer of the support, the free NiO is easily reducible, as well as Ni(OH)_2_, and these two species are preferentially formed on TiO_2_ as indicated by Spanou et al. [97].

In Figure 7c,d Nickel is shown dispersed in the same way as Ti, which suggests the interaction of Ni with Ti, promoting the formation of metallic nickel. The interaction between Ni and Ti and the reduction of Ti^4+^ to Ti^3+^ are related due to the reduction of Ni^2+^ to Ni^0^ [79]; the similarity of the micrographs, taking into account the distribution of the elements between Ti and Ni, implies that metallic nickel forms mostly on Ti; on Al it tends to mostly form a NiAl_2_O_4_ spinel. Escobar et al. [73] mentioned that the modification of the Ni/Al_2_O_3_ catalyst with TiO_2_ with 25% by weight, improves the distribution and reduction of the metal, and this could indicate that Ti can occupy vacancies in the Al_2_O_3_ network, which avoids the formation of NiAl_2_O_4_. This same effect was observed up to 0.1% of Ti in Co/Al_2_O_3_. The results obtained in the Ni/ATW1 A and Ni/ATW5 B micrographs propose that the reduction of Ni was more favorable over TiO_2_ and W because they inhibit the formation of the spinel phase.

In Figure 8a the micrographs of the Ni/ATW5 B catalyst are presented, as well as the results of the Ni/ATW1 A catalyst, suggesting a good dispersion of the elements that form the support (Al, Ti, and W). Nickel, on the other hand, Figure 7e, is homogeneously dispersed on the surface, and Ni nanoparticles are visibly in the lighter areas. In Figure 7e due to atomic contrast, nickel is preferentially dispersed over TiO_2_ and Al_2_O_3_. Nickel on the surface of the Ni/ATW5 B catalyst is lower compared with that of Ni/ATW1 A because of the diffusion of nickel within the pores of the support and the greater volume and pore diameter because of NH_4_OH and percentage of W. The parts of lower contrast are attributed to the support Al_2_O_3_-TiO_2_-WO_3_; in Figure 8b, the areas of high contrast, more agglomerated, suggest a strong interaction of Ni and Al due to the presence of NiAl_2_O_4_. As proposed by Liao et al. [96], Ni^2+^ species have a tendency to integrate inside of the Al_2_O_3_ structure.

#### 3.1.7. HRTEM

The results of the HRTEM micrographs of the monometallic catalysts Ni/ATW1 A and Ni/ATW5 A can be seen in Figure 9a–d and Figure 10a–d respectively. Figure 9a and Figure 10a show the monometallic catalyst Ni/ATW1 A and Ni/ATW5 A respectively, at a magnification of 200 nm. Dispersed particles can be observed, as well as agglomerated particles on the surface of the support.

In Figure 9b four rings were identified that correspond to the Ni phase with planes (111), (200), and (311) and NiAl_2_O_4_ with planes (220) having a polycrystalline material. The identification of NiAl_2_O_4_ in Figure 9b coincides with the XRD results, confirming the peak at 2θ = 65.5° that was associated with the phase of the spinel NiAl_2_O_4_ [70] overlapping with the signal at 2θ = 67° of the phase ƴ-Al_2_O_3_ due to non-stoichiometric spinel formation of NiAl_2_O_4_. In Figure 9c three phases were identified that correspond to NiO with an interplanar distance of 0.208 nm and 0.241 nm with planes indexed in (111) and (200) [98], Ni metallic with plane (111) with an interplanar distance of 0.203 nm and NiAl_2_O_4_ with plane (311) with an interplanar distance of 0.242 nm. The areas with high contrast correspond to the characteristic interplanar distance of Ni of 0.203 nm with plane (111), and the reduction of Ni is carried out in the particles with less NiO interaction.

For the Ni/ATW5 A catalyst, in Figure 10a,b, the planes (200) of the NiO structure and planes (200), (220), and (311) of Ni^0^ were identified, being a polycrystalline material as the catalyst Ni/ATW1 A. In Figure 10c three present phases were identified, with interplanar distances of 0.203 nm and 0.176 nm that correspond to the Ni structure of planes (111) and (200), respectively. Additionally, the NiO phase with an interplanar distance of 0.208 nm corresponds to plane (200) and NiAl_2_O_4_ at 0.201 nm corresponds to plane (400). The 0.176 nm distance was close to the interplanar distance of Ni (OH)_2_, so the 0.208 nm planes of NiO, which were located close to it, were due to the reduction of Ni(OH)_2_ during synthesis. Zhou et al. [99] stated that Ni formation occurs due to the reduction of NiO formed by the dehydration of Ni(OH)_2_ at a temperature of 400 °C, the nucleation of NiO is generated in different directions due to the polycrystalline characteristic of Ni(OH)_2_. Most of Ni diffuses into Al_2_O_3_, and occupying available sites forming NiAl_2_O_4_, W, and TiO_2_ could inhibit the growth of the NiAl_2_O_4_ spinel; therefore, we can observe that NiO is surrounded by NiAl_2_O_4_ at the Ni limit, so the Ni metallic nanoparticles originate mainly from NiO dispersed on the surface of the support with a weak interaction [100]. The particle size was affected by the percentage of W; an average size of 9.3 nm was obtained with 1%, and when increasing to 5% the particle size was 11.9 nm. The increase in particle size is due to the variation of the surface charge density of the species generated from W at pH 3 where the surface is positively charged by the species HW_6_O_21_^5−^ [101] generating larger particles with 5% of W having a greater interaction between NiO and W [35].

M.Mafokoane found [36] that the addition of WO_3_ changes the isoelectric point of the Al_2_O_3_ surface due to an increase in charge. They mentioned that the value of the isoelectric point of WO_3_ and Al_2_O_3_ was 4.2 and 7.7 respectively; however, it was concluded that at higher loads WO_3_(6%)-Al_2_O_3_ the isoelectric point presents a constant value, and this is because at higher loads of WO_3_ forms a monolayer on alumina at pH > 6–7. Cruz-Perez et al. [102] revealed that the isoelectric point of the Al_2_O_3_-TiO_2_ support is 6.0; however, this depended a lot on the preparation of the support. They described that at pH 4, adding W produces the ion W_12_O_4_0_8−_ where Ni can be coordinated tetrahedrally and octahedrally. On the other hand, for a solution of W at pH 9, the ion W_12_O_4_0_8−_ occurs, where Ni can be tetrahedrally coordinated. Even if the isoelectric point of Al_2_O_3_-TiO_2_ is 6, it causes polymerization from the WO_4_^−2^ monomer. According to the above, in the monometallic catalysts at pH 3, the Ni species with the increase in W can have a greater interaction with the support, coordinated tetrahedrally and octahedrally, as species NiAl_2_O_4_, which could cause a lower number of exposed metallic active sites, corroborating with the dispersion result seen in Table 4 and Figure 5 with TPD-H_2_.

Figure 11a shows dispersed particles and small agglomerations on the surface of the support. The diffraction analysis of Figure 11b reveals the presence of the diffraction rings corresponding to Ni (111) and (220). In Figure 11c, planes corresponding to Ni (111) and (200) with interplanar distance of 0.203 nm and 0.176 nm, NiO (200) at 0.208 nm, and NiAl_2_O_4_ (111) and (411) were identified. The MET micrographs of the Ni/ATW5 B catalyst showed the dispersed and isolated particles on the surface of the support, which can be seen in Figure 12a. According to the diffraction analysis of the electron select area shown in Figure 12b, the phases of NiO (200), NiAl_2_O_4_ (440), Ni (111), and (200) were identified. Figure 12c reveals how metallic Ni grew in different directions of the plane (111) and (200) due to the reduction of NiO coming from the dehydration of the polycrystalline structure of Ni(OH)_2_ that nucleates in different directions.

The 0.176 nm interplanar distance was adapted to the plane (200) of Ni, which indicates that the Ni originated from the NiO matrix [99] dispersed by the increase in the percentage of WO_3_, generating a better dispersion of nickel, and at the same time helping to inhibit the formation of the NiAl_2_O_4_ spinel; although Al_2_(WO_4_)_3_ was not identified in any catalyst, the XPS spectra, as discussed later, of the support suggest that there was an interaction of Al_2_O_3_ and WO_3_ due to the displacement of binding energies of the O1s. In Figure 11d and Figure 12d it is observed that as the percentage of W increased from 1% to 5%, the average crystallite size decreased from 10.2 nm to 8.5 nm. The modification of the support with W rises a better dispersion of Ni, contributing to a smaller particle size with 5% of W; the species of W at pH 9 could increase the negative charge on the surface of the support, improving the dispersion of the Ni^2+^ ions; the formation of Al_2_(WO_4_)_3_ can prevent the diffusion of Ni^2+^ into the Al_2_O_3_ network.

#### 3.1.8. FTIR-Pyridine

The results of the pyridine adsorption showed absorption bands of Lewis and Brönsted sites and are presented in the infrared region between 1700 and 1400 cm^−1^ corresponding to the absorbed pyridine [103].

Figure 13 shows the infrared spectra of all monometallic catalysts at 170 °C. The bands located at 1606 cm^−1^, 1575 cm^−1^, 1488 cm^−1^, and 1447 cm^−1^ characteristic to the absorption of pyridines in Lewis-type acid centers were identified, which corresponded to the interaction of pair electrons of pyridine on the metal cations. The bands at 1485, 1545, and 1640 cm^−1^ from the Brönsted sites generated a PyH* protonation on pyridinium ion [103,104]; however, none of these bands were detected. The bands located at 1445 cm^−1^ and 1606 cm^−1^ corresponded to the v19a and v8a coordination respectively; this is attributed to the coordinated pyridine in hydrogen bonds in Lewis acid centers generated by Ni metal sites and by the support [105,106].

Increased Lewis acidity may be generated because of the W=O bonds with unsaturated coordination of the W^6+^ species observed by both the Raman and FTIR spectra (not shown here). While, the bands at 1575 cm^−1^ with coordination v19a and 1488 cm^−1^ with coordination v8a correspond to weak Lewis’s acid centers [104]. As stated by Leal et al. [106] and Mafokoane et al. [36] the band at 1488 cm^−1^ is typical of pyridine adsorbed at sites (Lewis + Brönsted); the absence of Brönsted sites located at 1540 cm^−1^ and 1640 cm^−1^ decreased due to the Ni charge from the ion exchange of the Brönsted sites for the positive charge of the Ni^2+^ species, which were unsaturated, causing Lewis sites [50]. This is due to the Al_2_O_3_ coating on the support according to Raman spectra (not shown here); Tanabe et al. [107,108] mentioned that Al_2_O_3_ only has Lewis acid sites, and this Lewis acidity is related to Al^3+^ cations in tetrahedral coordination [109]. The absence of the 1540 cm^−1^ band indicated that there were no Brönsted sites on the surface strong enough to react with pyridine at a temperature of 200 °C. The low intensity observed in Figure 13 of the monometallic catalysts could be due to the coating of Ni metallic particles in Lewis-type active centers. These results are consistent with those presented by Jia et al. [76] where using Ni only promoted the presence of predominant Lewis sites on the surface of the monometallic catalyst.

The number of identified acidic sites were calculated by integrating the area under the curve. The results are shown in Table 4. The total number of Lewis sites was higher in the Ni/ATW B monometallic catalysts following this order: Ni/ATW1 B > Ni/ATW3 B > Ni/ATW5 B. The Ni/ATW1 B catalyst obtained higher Lewis acid sites. On the other hand, the Ni/ATW A catalysts preceded the following order of Lewis acid sites Ni/ATW3 A > Ni/ATW5 A > Ni/ATW1 A.

The increase in Lewis acid sites in the Ni/ATW3 A and Ni/ATW5 A catalyst is due to the low content of metallic Ni species, as a result of the incomplete reduction from the strong metal–support interaction because of the increase in W, generating the increase of Ni(OH)_2_; the Ni species of the Ni/ATW5 A catalyst were found in NiAl_2_O_4_ and Ni(OH)_2_ 2 as shown by the XPS spectra and the TPR profile. The increase in Lewis acid sites due to the increase in the W load could be caused by the increase in Ni(OH)_2_ concentrations with the increase in W. The increase in the number of sites is evident with the increase in surface area when comparing the Ni/ATW1 A catalyst with Ni/ATW5 B catalyst; however, it can be given that the W species generated on the support at different synthesis pH values [95]. The density of Lewis acid sites µmol/m^2^ in the Ni/ATW5 A catalyst was 0.89 µmol/m^2^, and it decreased with 5% of W due to the increase in the surface area with respect to the Ni/ATW1 A catalyst with an amount of acid sites of Lewis per m^2^ of 0.84 µmol/m^2^. It was observed that higher Lewis acid sites were generated at pH 9, possibly due to the species W_12_O_40_^8−^ and WO_4_^2−^ [95,101] increasing the density of Lewis acid sites.

#### 3.1.9. XPS

Figure 14 reveals Ni2p_3/2_ spectra of Ni/ATW catalysts. The broad Ni2p_3/2_ peak is associated with more than one nickel state. The presence of metallic nickel was identified in 852.4–852.6 eV and Ni^2+^ species, the latter presenting two types of interactions: one with the ƴ-Al_2_O_3_ and the other with the hydroxyl groups. The binding energy values as well as the percentage of the species are shown in Table 5. The existence of the Ni^3+^ species can be explained by the formation of a nickel spinel NiAl_2_O_4_ according to the energy values in 856.2–856.6 eV [110,111] as reported by Ruan and Zhang et al. [112,113]. On the other hand, the signal at 856 eV can be associated with the existence of Ni^3+^ species; therefore, the existence of three Ni species (Ni^3+^, Ni^2+^, and Ni^0^) can be seen. The interaction with the strong OH groups that remained on the surface after the reductive treatment resulted in the formation of nickel hydroxides Ni(OH)_2_ [114].

In Figure 14a–c it is observed how the Ni(OH)_2_ species grows with the content of W; a binding energy shift of 855.5 eV with 1% of W up to 855.2 eV is observed, and this shift of 0.3 eV is given by interaction of Ni with W particles, which was also observed in the TPR-H_2_ profiles, where reduction signals were detected at high temperatures with the increase of W to 5% *w*/*w*, suggesting the interaction of Ni particles with the W of the support [35]. It is possible that OH groups serve as a diffusion medium to interact within the aluminum matrix due to the wide peak at 856 eV, which may be due to the interaction of Ni^2+^ and Ni^3+^ species, occupying available tetrahedral and octahedral sites.

According to Q Zhang, the interaction of Ni^2+^ species with alumina favors two species: NiO (with oxygen from the network) and Ni(OH)_2_ (surface OH); Spanou et al. [97] mentioned that these species are formed mainly on the Ni/TiO_2_ surface and they are characterized by being mostly reducible. On the other hand, they are more difficult to reduce when carried out in abundant areas of the ƴ-Al_2_O_3_ phase, depending on the state of coordination being more difficult to reduce when they have octahedral coordination. The binding energies reported close to 856.5–857 eV correspond to Ni2p_3/2_ in the NiAl_2_O_4_ species [115]. Furthermore, the amplitude of the peak at 862 eV supports the presence of NiAl_2_O_4_, according to the literature on nickel aluminates; these species present an inverted spinel, where the divalent Ni^2+^ can occupy tetrahedral sites and the trivalent species, such as Ni^3+^, octahedral sites. It has been reported [74] that this species is formed at temperatures of 500 °C; based on XPS analysis and X-ray diffraction, it is inferred that free NiO forms a layer on the surface of the support, which is more easily to reduce unlike NiO that is bound to ƴ-Al_2_O_3_ sharing the same oxygen with aluminum.

Table 6 summarizes the values of the binding energies of the catalysts for Ti2p_2/3_, W4f_7/2_, Al2p, and O1s. In this table, slight changes in the binding energies are observed in the peak of W4f_7/2_ as the percentage of W, in the support increases. The change in binding energies suggests that both WO_3_ Al_2_O_3_ and TiO_2_ interacted by transferring electron density to WO_3_, causing a decrease in the binding energy of W4f.

The high-resolution Ti2p spectra of the Ni/ATW monometallic catalysts are shown in Figure 15. The broad peak at 458. eV, which could be attributed to the existence of 2 oxidation states, is in contrast to pure TiO_2_ according to the literature [116]. However, a broader peak is observed in the Ni/ATW5 A material, possibly due to the metal–support interaction considering the increase in the tungsten charge to 5%.

The Ti2p spectrum was analyzed using a Gaussian–Lawrencian curve with a shirley background where a main peak was found at 458 eV, presenting a FWHM value higher than 2 eV, which indicates the existence of not only the Ti^4+^ species. Two doublets located at 458.7 eV and 464.3 eV that belong to the Ti2p_3/2_-Ti2p_1/2_ Ti^4+^ species were identified; according to the binding energy difference of 5.6 eV, the Anatase phase is confirmed [117,118].

The Ti^4+^ reduction was possible due to the presence of Ni; the dissociatively chemisorbed hydrogen in Ni can diffuse from the Ni surface to the support where in areas of TiO_2_ can be reduced from Ti^4+^ to Ti^3+^, this effect being mentioned by Riyapan and Xu et al. [119,120] using Pd/TiO_2_ where the reduction of Ti^4+^ was observed at a temperature below 450 °C due to the homolytic dissociation of H_2_ on the support. This can generate the substitution of the W^6+^ ions in Ti^4+^ due to the close value of the ionic radius, as well as the interaction of W^6+^ modifying the chemical environment of Al^3+^, in its AlO_6_ and AlO_4_ coordination states in 74.6 eV and 73.5 eV as observed in Table 5.

All these factors contributed to the formation of Ti^3+^ species to compensate for charges generated by the elimination of hydroxyl groups, producing oxygen vacancies O^2−^, and these binding energies are observed in Figure 15 at values of 457.4 eV and 462.6 eV belonging to the Ti2p_3/2_ and Ti2p_1/2_ species for Ti^3+^. These charge transfers can leave deficient areas on the surface, originating unstabilized Tiᵟ^+^ species that can be in an electron transfer flux with Ni^3+^ species; the peaks found at 460 eV and 465.6 eV would correspond to the binding energy of Ti2p of Ni-Ti in the intermetallic state [121] with tetrahedral coordination [122].

The peaks located at 37.9 eV and 35.7 eV correspond to the binding energy Wf4_7/2_ and Wf4_5/2_ respectively, which are characteristic of the W^6+^ species [66]. The O1s peak of 531.5–531.6 eV is assigned to the OH and oxygen groups Al-O, easy to assign due to the difference in electronegativity of Al between the elements involved (Ti, W, Ni) and according to previous works [123]. As stated by Reddy et al. [124] there are O^2−^ vacancies generated during the synthesis method in a reducing atmosphere and in the ƴ-Al_2_O_3_ phase itself that help the dispersion of Ni^2+^ species because of these oxygen vacancies, where nickel coordinates at tetrahedral and/or octahedral sites. The peak at 530.5 eV belongs to the oxygen found in the network as a result of the contribution of the different elements, TiO_2_, WO_3_, and Al_2_O_3_ [117], according to the studies carried out by Benjaram M et al. [123]. The displacement to higher eV could be due to the chemical environment produced by the oxygen vacancies generated by H_2_ treatment; these defects generate a negative charge deficit that is compensated by the decrease in positive charge, that is, forming Ti^3+^ ions which actually act as electron donors, and in turn, new Ti-O-W bonds are generated with WO_3_ to stabilize the charges, demonstrating a decrease in the charge density on the Ti and Al atoms due to the bonding with WO_3_ [123] with charges of less than 1 and 3% by weight of W. The peak at 529.6 eV is caused by the interaction of Ni^2+^ in tetrahedral sites creating bonds of Ni-O [125] that combine with oxygens within the NiAl_2_O_4_ spinel because of this strong interaction.

### 3.2. Conversion of Levulinic Acid (LA) to γ-Valerolactone (GVL)

#### 3.2.1. Catalytic Activity of the ATW51 A

Figure 16 reveals the catalytic activity of support ATW1 A, which showed a conversion of 80% at 60 min of reaction; the conversion of LA increased until reaching 98% conversion in a reaction time of 240 min.

A 6% yield of GVL was obtained at 60 min. In general, the esterification of levulinic acid occurs in the presence of an alcohol such as methanol, ethanol, propanol, or butanol with an acid catalyst [7,32,126]. The conversion of levulinic acid of 79% in 1 h suggested the formation of methyl levulinate due to methanol as a reaction medium that favors the reaction [32,48,50]. Enumula et al. [34] studied a W-SBA-16 system with 3% *w*/*w* of W where the Lewis and Brönsted type acidity increased in the support and surface area, improving levulinic acid conversions. Selectivity to ethyl levulinate is higher with 3% W, and there is no improvement in catalytic activity using 5%.

##### ^1^H-NMR of the Crude after Reaction

Figure 17a shows the ^1^H-NMR spectrum of the ATW1A catalyst corresponding to the reaction crude after 2 h. There are strong signals of the esterification process of levulinic acid to methyl levulinate. The triplets generated by the neighboring CH_2_ protons were detected in their corresponding carbons (_#2_CH_2_, δ = 2.57 ppm and _#3_CH_2_, δ = 2.76 ppm); additionally, the singlets for these protons (_#5_CH_3_, δ = 2.19 ppm and _#6_CH_3_, δ = 3.67 ppm) were noticed. Figure 17b shows the ^1^H-NMR spectrum of a sample analyzed after 4 h of reaction. Signals of the protons of the conversion of Levulinic Acid were recognized (singlet _#5_CH_3_, the triplets of the protons _#2_CH_2_ and _#3_CH_2_), methyl levulinate or methyl 4-oxopenanoate (singlets of the protons _#5_CH_3_ and _#6_CH_3_, the triplets of _#2_CH_2_ and _#3_CH_2_ protons), Methyl 4-Hydroxypenanoate (_#5_CH_3_, _#6_CH_3_ protons singlets, _#2_CH_2_,_#3_CH_2_ protons triplets and a _#4_CH proton signal multiplet), and γ-Valerolactone generated the multiplicity of neighboring protons for #3CH_2_ and #4CH_2_ multiplicity of #5CH, and finally the doublet corresponding to #6CH_3_ developed from its neighbor CH. These results are very important because the conversion of levulinic acid into reaction by-products such as esters can be monitored, where acidic sites are needed and, in the hydrogenation process, it is required to have available active metal sites on the catalyst surface to produce γ-Valerolactone. 

### 3.3. Catalytic Tests of the Ni/ATW Catalysts

Figure 18 shows the results of the monometallic catalysts at different times. The yields decrease when the % of W increases, and this is due to: (a) the decomposition of levulinic acid by the Ti^3+^ species generating a carbon deposit on the surface resulting in lower yields or (b) by the adsorption of water molecules released by esterification or by physisorbed water from the support. Pham et al. [127] mentioned that the Ti^3+^ species cause the decomposition of levulinic acid because the ketone group of levulinic acid binds preferentially to these Ti sites and they are coordinately unsaturated through the carbonyl oxygen atom, so despite from the high conversion of levulinic acid, the yield of GVL decreases; these results are similar to those presented by the group of Mafokoane et al. [36].

On the other hand, the 24 h GVL yield of the Ni/ATW B catalysts in Figure 19 showed the following trend: Ni/ATW5 B > Ni/ATW3 B > Ni/ATW1 B (see Table 7). From the results of the IR spectra of pyridine it can be observed that, while the amount of Lewis sites decreased, the yield of GVL increased; in addition, the increase in surface area causes a better dispersion of the Ni particles, and this indicates that the average pore diameter increased due to the fact that the ATW support was modified at pH 9 using NH_4_OH, which could have helped the Ni deposit within the pores because the W species generated dispersed more efficiently the Ni in the support at pH 9 with NH_4_OH, and it was also observed that as the W load increased, the GVL performance increased.

#### 3.3.1. ^1^H-NMR Elucidation by Products

For this Ni/ATW1 A nickel monometallic catalyst (See Figure 20), methyl levulinate signals were elucidated by finding the characteristic triplets generated by the vicinal CH_2_ with their displacements for each proton (#2CH_2_, δ = 2.55 ppm and #3CH_2_, δ = 2.78 ppm); a singlet was observed for the methyls (#5CH_3_, δ = 2.19 ppm) and (#6CH_3_, δ = 3.70 ppm). In the case of the GVL, the multiplicity of the neighboring protons were generated for (#4CH_2_ δ = 1.88 ppm and 2.38 ppm) and (#3CH_2_ δ = 2.45 ppm), the multiplicity of (CH, δ = 4.66 ppm), and finally the doublet (CH_3_, δ = 1.41 ppm) generated from its neighbor CH [128]. The strong signals for methyl levulinate correspond to the excess percentage.

The different reaction routes are shown in Figure 21. GVL’s poor performance is due to competition from sites; Lewis sites were generated by the support and by Ni^2+^ species in the form of NiO, Ni(OH)_2_ and NiAl_2_O_4_, first promoting esterification, forming Methyl-Levulinate, and subsequently being hydrogenated by metallic sites. At the same time, according to the results obtained by ^1^H-NMR, LA is hydrogenated to form 4-hydroxypentanoic acid. The existence of Lewis sites that provides the catalyst monometallic catalysts promotes the esterification of LA to methyl-levulinate; simultaneously, LA is hydrogenated to 4-hydroxypentanoic acid due to the metallic sites, and subsequently GVL is obtained. However, considering the medium of reaction in methanol and the competition of Lewis sites, the esterification of Levulinic Acid (LA) and the conversion of 4-hydroxypentanoic acid to GVL need to be minimal in the first hours of reaction.

Therefore, at 24 h when 100% conversion of Levulinic Acid has been reached, there is no competition from metallic sites for the hydrogenation of Levulinic Acid; therefore, the hydrogenation of methyl levulinate is more selective to form methyl 4-hydroxypentanoate so that later dehydration by Lewis sites can take place and finally intramolecular dealcoholization, forming γ-Valerolactone.

#### 3.3.2. Effect of Lewis Acid Sites on the Yield to GVL

The increase in Lewis acid sites (see Table 4) did not affect the conversion of Levulinic Acid for the Ni/ATW1 A catalysts; for Ni/ATW3 B it was 100% and 97% for the Ni/ATW5 A catalyst (see Figure 22 and Table 7). The maximum conversion of GVL was obtained with 1% of WO_3_ of the Ni/ATW1 A catalyst with 59% (see Table 7) with 207.9 µmol/g_cat_. When 1% increases to 3% the Lewis acid sites increased up to 428.9 µmol/g_cat_, decreasing the GVL yield up to 27% (see Table 7), and the conversion to GVL lowered the selectivity with the increase in the Lewis acid sites. It is expected that the Ni/ATW5 A catalyst with 246 µmol/g_cat_ increased the GVL yield; however, a GVL yield of 18% (see Table 7) was obtained, and this could be due to the increase in surface area and the increase in Ti^3+^ species because the increase in 5% of WO_3_. Although the yields decreased with the increase of WO_3_ in the support, the reaction was not selective to α-angelica lactone, but rather followed the route of the esterification of Levulinic Acid. These results suggest that with an excess of Lewis acid sites the yield of GVL decreases, generating the decomposition of Levulinic Acid and limiting the conversion to methyl levulinate, lowering the selectivity to GVL. For Ni/ATW B catalysts, the maximum GVL yield was obtained with 5% of WO_3_ and Ni/ATW5 B catalyst with 80% with 385.9 µmol/g_cat_. The Lewis acid sites increased to 445.9 µmol/g_cat_ with 3% of WO_3_, decreasing the GVL yield to 65% for the catalyst with 1% of WO_3_ the amount of Lewis acid sites was 458.9 µmol/g_cat_; under this system, the maximum acidity was found using only 1% of WO_3_, though it is clear that the conversion to GVL decreases with the increase in Lewis acid sites. It was expected that the GVL performance of Ni/ATW1 B catalyst would be like the Ni/ATW3 B catalyst since the number of basic sites is alike, but it did not follow that trend, and this may be due to the WO_3_ species that were found on the surface, generating dispersion of the Ni nanoparticles. The effect of NH_4_OH influences the Ni deposition and the OH groups favor the dispersion of Ni, improving the yield of GVL.

#### 3.3.3. Effect of the Ni Particle Size on the Yield to GVL

In Figure 23a, it is observed how the particle size influences the GVL yield after 24 h of reaction. A yield of 59% of GVL was obtained when the particle size was 9.2 nm (TEM, see Table 4) in the Ni/ATW1 A catalyst, and the formation of small particles with a low W load (1% *w*/*w*) was favored; with 5% *w*/*w* of W, the particle size increased to 11.9 nm (TEM, see Table 4), affecting the GVL yield by only obtaining 18% in the Ni/ATW5 A catalyst. These results show the influence of the particle size in the selectivity to GVL, the low nickel content helping in the generation of particles smaller than 10 nm [46]; however, the dispersion and the interaction with the support can modify the particle size due to a greater metal–support interaction [35,43]. It was studied how the particle size, the percentage of nickel, and the amount of Lewis acid sites show a favorable synergistic effect to have a yield at GVL. According to the results, when there is a high percentage of nickel, the particle size is greater than 10 nm, affecting the performance at GVL. Small nickel particles showed higher activity due to higher Ni surface area or due to higher reactivity of less coordinated sites. It is interesting to note that when the particle size was smaller with 1% W in the Ni/ATW1 A catalyst, there was a greater amount of NiAl_2_O_4_, which suggests the existence of Ni^2+^ and Ni^3+^ species acting as Lewis sites, responsible for levulinic acid adsorption and the activation of the carbonyl group of the carboxylic acid in the small particles of metallic nickel [34,131]. As reported by Song [132], the adsorption of H_2_ would take place on Ni (111) and levulinic acid on NiO (111). Diffusion of H_2_ species occurs through hydroxyl groups [133]; the synergy of support and metal sites is necessary for efficient hydrogenation of levulinic acid to GVL.

The particle size obtained by TEM is compared with the yield of GVL in Figure 23b. The Ni/ATW5 B catalyst showed a higher performance, the improvement in these results being caused by the particle size of 8.5 nm (TEM, see Table 4) compared with the Ni/ATW1 B catalyst because the spillover effect is carried out in small particles, dissociating the H_2_. The presence of OH groups is important in this type of reaction, since they function as H_2_ transport, so the Ni(OH)_2_ species as well as M-OH groups in the support serve as a diffusion medium. Cheng et al. [134] reported that it is likely that this effect depends equally on the acidity of the support and the synthesis method of the catalyst. Meith et al. [135] mentioned that when a pH of 5 is used, the predominant species are Ni^2+^, however at a pH greater than 6.5 these species transform into Ni(OH)_2_, increasing the complexity of the system. Therefore, in the support at pH 9, the surface OH groups of the support generate a better dispersion of the Ni^2+^ species when they exchange with the OH groups forming Ni(OH)_2_, and these species in presence of molecular H_2_ due to the spillover effect can reduce nickel in situ by increasing metallic sites. When the OH groups decrease, the nickel begins to nucleate on and around the already fixed metal sites [136], growing the particle size and diminishing the yields of GVL.

#### 3.3.4. Effect of de Ni^0^ Metallic Sites on the Yield to GVL

The results obtained in this research work suggested that the yield towards GVL decreases with an excess of Lewis acid sites, generating the decomposition of levulinic acid limiting the conversion to methyl levulinate, lowering the yield to GVL. However, the yield to GVL did not increase for the Ni/ATW5 A catalyst, although the Lewis acid sites were as the Ni/ATW1 A catalyst. This suggests that a necessary contribution of metallic sites with the Lewis acid sites is needed to have a high yield, the low yield of GVL is not only a function of the Lewis acid sites, but also of the percentage of metallic Ni. Figure 24 shows how the GVL performance is a function of the percentage of metallic Ni. Since the Ni/ATW1 A catalyst contains more metal sites 9.1%, H_2_ can be dissociated in a homolytic way more effectively and by spillover effect, and H_2_ is transported by OH groups [79,133]. Nonetheless, it is possible that when diffusing in the support it reduces NiO in situ, Ni(OH)_2_ and NiAl_2_O_4_ species close to metallic Ni particles. In the TEM images, the interplanar distances of NiO, Ni(OH)_2_, and NiAl_2_O_4_ within the limits of metallic Ni were observed, which would lead to a consumption of H_2_ due to the reduction of these species, limiting the hydrogenation of levulinic acid or methyl levulinate, obtaining low yields in the first hours of reaction and a maximum yield at 24 h of reaction because of the generation of new active sites by the spillover effect.

## 4. Conclusions

The aim of this work was the modification of the Al_2_O_3_-TiO_2_-WO_3_ support using different pH of synthesis with different percentages of W and Ni at 10% by weight by the suspension method and its application in obtaining GVL from LA using methanol as solvent. The nickel particles dispersed adequately in the support, and the surface area as well as the volume and the pore diameter decreased once the nickel was deposited. All the catalysts showed a complete conversion of LA, this due to the esterification to methyl levulinate according to the results of ^1^H-NMR. The high selectivity of esterification was attributed to the Lewis acid sites that all catalysts exhibited. The low percentage of metallic Ni could indicate the contribution of Lewis sites due to the presence of NiAl_2_O_4_ and Ni(OH)_2_ species. Among all the catalysts prepared, the Ni/ATW5 B catalyst at pH 9 using 10% Ni and 5% *w*/*w* of W, in the support, showed the highest yield towards GVL, 80% at 24 h of reaction and 4 MPa of H_2_ pressure; this suggests, according to TEM studies, that the catalytic activity is optimal at a smaller particle size. This was related to the W species generated at different pH of synthesis on the support that influenced the Ni particle size. The yield to GVL increased as the W amount in the support increased. These results suggest that the increase in W, in the support at pH 9, improves the stability of the Ni metallic particles, which leads to electronic and structural changes between the active phase (Ni metallic) and the Lewis acid sites, generating esterification from LA to methyl levulinate and increasing the yield to GVL by the spillover effect. However, when the Ni catalyst is supported on a modified support at pH 3, the generation of the species reduction from Ti^4+^ to Ti^3+^ is promoted, causing a decrease in the yield to GVL.

## Data Availability

Not applicable.

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
