# Peer review of "γ-Valerolactone Production from Levulinic Acid Hydrogenation Using Ni Supported Nanoparticles: Influence of Tungsten Loading and pH of Synthesis"

_nanomaterials, 2022, doi:10.3390/nano12122017_

Round 1

Reviewer 1 Report

The authors conducted a study on the role of Ni particles in Al2O3-TiO2-W supports to produce Y-GVL from Levulinic acid. Different pHs during the synthesis of supports induced different W species that affect Ni particles, resulting in a better yield of Y-GVL, which the authors comprehensively characterize and analyze the catalyst and liquid product. One minor suggestion is that further discussion of the expected position and dispersion of Ni particles in terms of points to zero charges of differently synthesized supports would likely increase the understanding of the paper. This paper deserves publication in nanomaterials when the above is taken into account. 

Author Response

Villahermosa Tab., México; June 06, 2022

Dear Reviewer, 1:

Thank a lot for their recommendations. They were an excellent guideline to improve the manuscript.

Open Review

The authors conducted a study on the role of Ni particles in Al2O3-TiO2-W supports to produce Y-GVL from Levulinic acid. Different pHs during the synthesis of supports induced different W species that affect Ni particles, resulting in a better yield of Y-GVL, which the authors comprehensively characterize and analyze the catalyst and liquid product.

Question 1) One minor suggestion is that further discussion of the expected position and dispersion of Ni particles in terms of points to zero charges of differently synthesized supports would likely increase the understanding of the paper. This paper deserves publication in nanomaterials when the above is taken into account. 

Response 1): We improved the discussion of the expected position and dispersion of Ni particles in terms of points to zero charges. First, we added section 3.1.5 Temperature-Programmed Desorption of Hydrogen (TPD-H2) in order to explained more about species of Ni cero valent and the metallic dispersion values of Ni/ATW A and Ni/ATW B. In the section 3.1.4 TPR-H2 is explained the process of reduction of NiO to Ni0 in our synthesized nanocatalysts. In the section 3.1.3 is explained the presence of Ni species in our synthesized nanocatalysts, like in octahedral and tetrahedral coordination. Besides, we discussed further about species of Ni on line 628 to 641, using previous studies in this field.

NOTE: Additionally, we corrected several typos error and rewritten some paragraphs and sentences.

We indicated all changes in the manuscript with red letter.

Sincerely Yours

The authors

Reviewer 2 Report

The manuscript “γ-Valerolactone production from Levulinic acid hydrogenation using Ni supported nanoparticles: Influence of Tungsten loading and pH of synthesis” reports a very accurate investigation of γ-Valerolactone synthesis and a complete characterization of the studied materials.

The paper is well written, however in some points it is too verbose, and I would recommend using a more schematic style to facilitate the reader.

In my opinion the manuscript is suitable for publication in Nanomaterials journal after a very minor revision to increase the reader’s interest. I do not require any experimental or scientific revision; however, the following suggestions are proposed:

Title

Please pay attention to the Greek letter gamma (γ instead of y or Y).

The same typo is also in the abstract.

Abstract

In the abstract session, the Authors should replace the symbol W with the full name of the chemical element Tungsten the first time it appears in the text.

Introduction:

In the introduction session is very important the comparison with other research in this field or previous studies, indeed, as reported by the references, the authors possess a good and consolidated knowledge of the subject treated here.

However, the novelty and significance of the present work should be emphasised.  

In my opinion, the synthesis of the GVL reported by other authors should be detailed in another paragraph and not in the introduction. In addition, it would also be better to resume the different procedures in a summary table including all information (catalysts and its %, temperature, pH, solvent…).

Author Response

Villahermosa Tab., México; June 06, 2022

Dear Reviewer, 2:

Thank a lot for their recommendations. They were an excellent guideline to improve the manuscript.

Open Review

The manuscript “γ-Valerolactone production from Levulinic acid hydrogenation using Ni supported nanoparticles: Influence of Tungsten loading and pH of synthesis” reports a very accurate investigation of γ-Valerolactone synthesis and a complete characterization of the studied materials.

The paper is well written, however in some points it is too verbose, and I would recommend using a more schematic style to facilitate the reader.

In my opinion the manuscript is suitable for publication in Nanomaterials journal after a very minor revision to increase the reader’s interest. I do not require any experimental or scientific revision; however, the following suggestions are proposed:

Question 1) Title

Please pay attention to the Greek letter gamma (γ instead of y or Y).

The same typo is also in the abstract.

Response 1): We fixed this typo error in the title and the abstract and also in the whole manuscript.

Question 2) Abstract

In the abstract session, the Authors should replace the symbol W with the full name of the chemical element Tungsten the first time it appears in the text.

Response 2): We replaced the symbol W with the full name of the chemical element Tungsten the first time it appears in the text.

Question 3) Introduction:

In the introduction session is very important the comparison with other research in this field or previous studies, indeed, as reported by the references, the authors possess a good and consolidated knowledge of the subject treated here.

However, the novelty and significance of the present work should be emphasised.  

Response 3): We improved the discussion in the introduction section, we added another previous studies in this field besides they had already before in the manuscript. It can be seen on line 123 to 131, 142 to 144, 167 to 170, 180 to 187. We emphasized novelty and significance of our work; it can be seen on line 171 to 179.

Question 4) Introduction:

In my opinion, the synthesis of the GVL reported by other authors should be detailed in another paragraph and not in the introduction. In addition, it would also be better to resume the different procedures in a summary table including all information (catalysts and its %, temperature, pH, solvent…).

Response 4): The synthesis of the GVL and mechanism reaction were described deeply in the manuscript and we made a resume of experimental conditions reported by other authors in the Table 1 in order to clarify the discussion.

We indicated all changes in the manuscript with red letter.

Sincerely Yours

The authors
